# Dietary diversity and association with non-communicable diseases (NCDs) among adult men (15–54 years): A cross-sectional study using National Family and Health Survey, India

Mriganka Dolui[1], Sanjit Sarkar[1]*, Pritam Ghosh[2,3], Moslem Hossain[1]

1 Department of Geography, School of Earth Sciences, Central University of Karnataka, Karnataka, India,
2 Department of Geography, University of Calcutta, Kolkata, West Bengal, India, 3 Department of
Geography, Ramsaday College, Howrah, West Bengal, India

* sanjitiips@gmail.com

University, INDIA

**Data Availability Statement:** The dataset analyzed
during the current study are available in the
Demographic and Health Survey (DHS) repository

## Abstract

A healthy and diversified diet is essential for preventing several non-communicable dis-
eases (NCDs). Given the increasing evidence of diet-related health burdens and the rising
prevalence of NCDs among Indian adults, the present study aims to explore dietary diversity
patterns among adult men in India and their association with non-communicable diseases
(NCDs). For this purpose, the study used the fourth round of the National Family and Health
Survey (NFHS-4) to analyze adult male samples (n = 1,12,122). Dietary Diversity Scores
(DDS) were computed by the weighted sum of the number of different food groups con-
sumed by an individual. The prevalence of diabetes, heart disease, and cancer among adult
men is considered a non-communicable disease. Bivariate and logistic regression was car-
ried out to examine the association between DDS and NCDs by estimating chi-squared
tests ($\chi^2$-test), odds ratio (OR), and 95% confidence interval (CI). The prevalence of diabe-
tes, heart disease, and cancer among adult men in India is 2.1 percent, 1.2 percent, and 0.3
percent, respectively. Results show a positive association between dietary diversity score
and the prevalence of the non-communicable disease. High-level dietary diversity scores
increase to two times the likelihood of diabetes (OR 2.15 with p<0.05) among adult men
than to better-off counterparts while controlling all the covariates. However, a moderate die-
tary diversity score significantly decreases the likelihood of heart disease (OR 0.88 with
p<0.10) and Cancer (OR 0.71 with p<0.05) for adult men compared to a lower score of die-
tary diversity. In addition, age, marital status, drinking and smoking habits, occupation, and
wealth index are also significantly associated with the odds of non-communicable diseases
among adult men.

## 1. Introduction

Globally, Noncommunicable Diseases (NCDs) accounted for 74 percent of total death [1] and
emerged as the primary cause of premature mortality. Cardiovascular diseases are the leading

at https://dhsprogram.com/data/available-datasets.cfm, and can be accessed on formal request.

**Funding:** The authors received no specific funding for this work.

**Competing interests:** The authors have declared that no competing interests exist.

cause of most NCD deaths worldwide. Indeed, three-quarters of global NCD deaths occur in low- and middle-income countries [2]. A developing country, like India, is also showing a rising burden of non-communicable diseases (NCDs), where NCDs induce more than 60 percent of all deaths [3]. Correspondingly, the prevalence of various life-threatening non-communicable diseases also shows a rising trend in India for both men and women between 2015–16 and 2019–21 [4]. The prevalence of diabetes shot up from eight to 16 percent and six to 14 percent, respectively, for adult men and women in India during the last five years. A similar trend is also reported for hypertension in both the sexes during the same period. Although, diabetes and hypertension are not the direct cause of mortality but they boost other causes of mortality. However, cardiovascular diseases, such as heart disease, are reported as the most significant cause of death that caused 30 percent of all deaths and 50 percent of all NCDs-related deaths globally [5]. Studies showed that poor, unhealthy diets and lack of physical exercise are the major causes of the population's burden of heart diseases, diabetes, hypertension, etc. [6,7].

Consumption of diversified foods leads to healthy life of a human being, which may also prevent many non-communicable diseases [8–13]. Nonetheless, the relationship between dietary diversity and non-communicable diseases is dynamic and depends on the consumption of specific types of food groups. Access to more diversified diets can lead to higher fats, resulting in other health problems [14]. Optimum intake of micronutrients such as vitamins C and E may reduce the risk of heart and cardiovascular diseases [15–17]. Therefore, a balanced diet that includes an essential quantity of nutritional components such as calories, protein, fat, micronutrients, and dietary fibers is required for long-term health and well-being. Some food groups are rich in specific nutrients content; thus, a diversified food basket may enrich the households' nutrient adequacy and nutritional status [10,17–20]. For example, the intake of whole grains improves diet quality and is associated with beneficial health outcomes [21]; meats [22] and fish are also major sources of protein and fat [22,23]. Similarly, vegetables, milk-dairy products, and fruits are the major sources of several nutrients and micronutrients [24].

Several studies have used Dietary Diversity Scores (DDS) to assess the diet diversity pattern in households or individual levels [25,26]. Most of the studies have calculated DDS based on a 24-hours recall period by summing the number of food groups consumed on the last day. Although the 24-hour recall method is subject to less-recall bias, it has a quality disadvantage because it does not incorporate the food consumption frequencies which is also important to understand diet diversity patterns. Therefore, in this study, we adopted 30 days recall method where consumptions of specific food groups and their frequencies have also been considered. Though, many studies [27–34] have established the linkage between diet diversity and health adversity such as malnutrition, anemia, and cardiovascular diseases among various population sub-groups or in general. But studies on the relationship between diet diversity and NCDs in Indian adult men are limited. Therefore, the present study examines the regional variations of dietary diversity and its association with non-communicable diseases among adult men in India.

## 2. Methodology

### 2.1 Data and sample

Data for this analysis was obtained from the fourth round of the National Family and Health Survey (NFHS-4, 2015–16). The NFHS is an Indian version of the Demographic and Health Survey (DHS). The survey was conducted by the International Institute for Population Sciences (IIPS) with the stewardship of the Ministry of Health and Family Welfare (MoHFW), Govt. of India (GOI), and technical support from ICF international. A two-stage sampling

procedure was followed in the NFHS-4. In rural areas, villages are selected as Primary Sampling Units (PSUs) in the first stage, followed by a random selection of households in each PSU in the second stage. Similarly, in urban areas, Census Enumeration Blocks (CEBs) are selected in the first stage and a random selection of households in each CEB in the second stage. However, the households at the second stage in both rural and urban areas were selected after conducting a complete mapping and house listing in the selected first-stage units. The NFHS-4 provides information on diet diversity, health well-being, current morbidities, socio-economic and demographic characteristics of adult men across the districts and states/UT in India, along with other necessary information. NFHS-4 collected information from 6,01,509 households, 6,99,686 women (15–49 years), and 1,12,122 men (15–54 years). Thus, considering the study's objective, we have restricted our analysis to the men samples only.

## 2.2 Outcome variables

The prevalence of non-communicable diseases such as diabetes, coronary heart disease, and cancer among adult men are considered outcome variables for this study. The outcome variables are codded as binary. If any adult man who reported any of these morbidities is assigned as 'yes', else 'no'.

## 2.3 Dietary diversity score

The dietary diversity of adult men is considered an independent variable. Dietary information and consumption frequencies (daily, weekly, occasionally, or never) were collected for nine food groups within the last month. The food groups included in the survey, are as follows: (i) milk or curd, (ii) pulses or beans, (iii) dark green leafy vegetables, (iv) fruits, (v) eggs, (vi) fish, (vii) chicken or meat, (viii) fried foods, (ix) aerated drinks. A dietary diversity score (DDS) was calculated by the weighted sum of the number of different food groups consumed by an individual in the last month. The food groups were weighted by level of consumption frequencies. The weightage values assigned for different consumption frequencies are zero for never consumed; one for occasionally consumed; two for weekly consumed; and four for daily consumed, representing lower to the higher intensity of diet diversity. Thus, the weighted sum of all food groups ranges from '0' to '36'. The score value '0' means not a single food group was consumed ever during the reference period, whereas the score value '36' means all the nine food groups were consumed 'daily' by the individual during the reference period. The dietary diversity score (DDS) was classified as low ($\leq 12$), moderate (13–25), and high ($\geq 26$) for further analysis.

## 2.4 Covariates

A set of covariates are also included in the study, possibly related to different non-communicable diseases. The covariates in the analysis include age (15–29, 30–44, 45–54 years), marital status (unmarried, married, and separated), education (no education, primary, secondary, and higher), types of occupation (no occupation, agricultural. Services, skilled and unskilled, and others), smoking (yes/ no), drinks alcohol (yes/no), religion (Hindu, Muslim, and Others), castes (Scheduled caste, scheduled tribe, other backward class, and general), residence (urban/ rural), wealth index (poorest, poorer, middle, richer, and richest), region (north, central, east, northeast, west, and south).

## 2.5 Statistical approach

Bivariate analyses were carried out to show the sample distribution, the prevalence of non-communicable diseases, and dietary diversity among adult men by their background variables.

Further, to test the significance of the association between dietary diversity and non-communicable diseases among adult men, chi-squared tests ($\chi^2$-test) were performed. Further, GIS-based mapping tools were used to outline the geospatial pattern of dietary diversity score and the prevalence of diabetes, coronary heart disease, and Cancer among Indian adult men. Multivariable binary logistic regressions were applied to find out the odds of occurrence of the NCDs and significant risk factors associated with the dietary diversity among adult men. The results of regression analyses were interpreted as an odds ratio (OR) with 95% confidence interval (CI). The mathematical function of binary logistic regression used in this study is as follows:

$$\ln\left(\frac{p_i}{1-p_i}\right) = \alpha + \chi_1\beta_1 + \chi_2\beta_2 + \chi_k\beta_k + \epsilon$$

Where, $\ln\left(\frac{p_i}{1-p_i}\right)$ is the odds in which $p_i$ is the probability of 'i' individual experiencing with the outcome events, $\alpha$ is the constant, $\chi_i$ is the vector of the predictor variables, $\beta_i$ is the vector of regression coefficients, and $\epsilon$ is the unexplained part or error term.

## 3. Results

### 3.1 Sample background characteristics

Table 1 represents the background characteristics of the adult male population selected for the study. Results show that the mean age of the sample population is 32 years, and 46 percent of the sample population belongs to the 15–29 years of age category. More than half of the sample population (63 percent) are married; 82 percent belong to Hindu, whereas 44 percent and 28 percent of the adult male are from other backward classes (OBC) and general cate, respectively. About 62 percent of the total men population resides in rural areas, half of the sample (63) have qualified for their secondary education, and 17 percent have done higher education. Regarding occupational status, nearly 27 percent of the sample population is in agriculture, and one-fourth of the total population is working as skilled and unskilled manual workers.

### 3.2. Dietary diversity patterns

Fig 1 depicts the food consumption pattern among adult men in India. Milk/curd, pulses/beans, and vegetables are the most common food groups consumed by adult men. However, the consumption of protein-rich foods daily, such as eggs, fish, and chicken, is significantly lower (< 5 percent). Even only half of the adults (<45 percent) have consumed such protein-rich foods weekly. Dietary diversity also varies significantly across geographical regions in India (Fig 2A). Mean dietary diversity scores are lower in the western part of India, especially in Rajasthan, Gujrat, and Madhya Pradesh districts. Contrarily, the districts of Karnataka, Kerala, Tamil Nadu, West Bengal, and Odisha have reported higher mean dietary diversity scores.

### 3.3. Prevalence of non-communicable diseases (NCDs)

Nearly 2.1 percent, 1.2 percent, and 0.3 percent of adult men in India suffered from major non-communicable diseases, such as, diabetes, coronary heart disease, and cancer, respectively (Table 2). The prevalence of NCDs varies across socio-demographic characteristics. There is a significant increase in the prevalence of diabetes, coronary heart disease, and cancer with the increment in age. Adults separated from their marital relationship suffer more from diabetes (3.2 percent) and coronary heart disease (2.3 percent). Similarly, the prevalence of diabetes and coronary heart disease are highest among adult men with higher education (2.5 percent

**Table 1. Background characteristics of the adult men (15–54 years) selected for the study (N = 112122).**

| Background Characteristics | Percentage | N |
|---|---|---|
| Age | | |
| 15–29 | 45.96 | 51535 |
| 30–44 | 36.05 | 40425 |
| 45–54 | 17.98 | 20162 |
| Mean age (years) | 32 | 112122 |
| Marital Status | | |
| Unmarried | 35.46 | 39762 |
| Married | 63.13 | 70781 |
| Separated/Widow/Divorced | 1.41 | 1578 |
| Education | | |
| No Education | 13.01 | 14590 |
| Primary | 12.57 | 14091 |
| Secondary | 57.09 | 64009 |
| Higher | 17.33 | 19431 |
| Occupation | | |
| No occupation | 21.96 | 24623 |
| Agricultural | 26.94 | 30202 |
| services | 7.13 | 7991 |
| Skilled and unskilled manual | 25.79 | 28911 |
| Others | 18.19 | 20394 |
| Smoking | | |
| No | 73.61 | 82531 |
| Yes | 26.39 | 29591 |
| Drinks alcohol | | |
| No | 70.49 | 79036 |
| Yes | 29.51 | 33086 |
| Religion | | |
| Hindu | 81.51 | 91390 |
| Muslim | 13.19 | 14789 |
| Others | 5.30 | 5942 |
| Caste | | |
| SC | 19.75 | 22139 |
| ST | 8.80 | 9871 |
| OBC | 43.56 | 48841 |
| General | 27.89 | 31270 |
| Residence | | |
| Urban | 38.31 | 42953 |
| Rural | 61.69 | 69169 |
| Wealth Index | | |
| Poorest | 14.66 | 16440 |
| Poorer | 18.64 | 20904 |
| Middle | 21.13 | 23687 |
| Richer | 22.28 | 24976 |
| Richest | 23.29 | 26114 |
| Region | | |
| North | 15.21 | 17054 |
| Central | 0.49 | 545 |

*(Continued)*

**Table 1.** (Continued)

| Background Characteristics | Percentage | N |
|---|---|---|
| East | 16.41 | 18401 |
| Northeast | 23.03 | 25819 |
| West | 4.83 | 5416 |
| South | 40.03 | 44887 |
| Total | 100.00 | 112122 |

Note: SC-Scheduled caste; ST-Schedule tribes; OBC-Other backward cast.

and 9 percent, respectively) and adults in service (3 percent and 1.4 percent, respectively), who have smoking and drinking habits. Regarding caste, General caste people suffered most from diabetes, whereas coronary heart disease and cancer prevalences were highest among Scheduled Tribes (1.5 percent) and Other Backward Caste (0.35 percent) categories, respectively. The adult men who reside in urban areas and belong to the higher economic class have reported the highest prevalence of diabetes than their counterparts. In terms of geographical variations in India (Fig 2B–2D), dieabetes are prevalent in southern, eastern-costal regions whereas heart diseases and cancer are mostly scattred in southern, central and eastern regions.

### 3.4. Associations between non-communicable diseases (NCDs) and dietary diversity patterns

Table 3 shows the prevalence of non-communicable diseases among adult men by food consumption patterns. The analyses show that the prevalences of non-communicable diseases vary with the consumption frequencies of selected food groups. The analyses show that the prevalence of diabetes is higher among those who consume milk /curd, pulses/beans, dark green leafy vegetables, and fruits daily or never. On the other hand, those who never consumed milk /curd, pulses/beans, dark green leafy vegetables, and fruits in the last month have reported the highest prevalence of coronary heart disease and cancer. However, daily consumption of protein-rich foods such as eggs, fish, and chicken shows significantly higher associations with all three non-communicable diseases (p <0.05). Similarly, a significant association was established in Table 4 between dietary diversity score and non-communicable

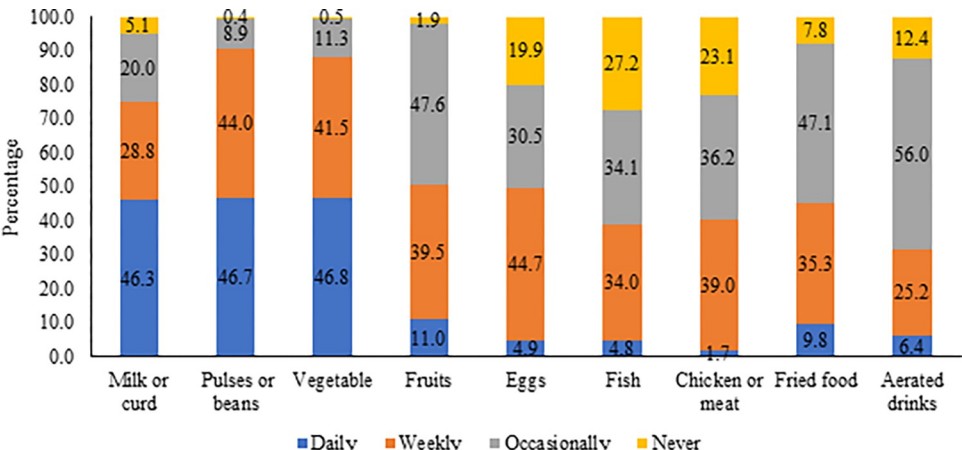

**Fig 1. Food consumption patterns among adult men in India.**

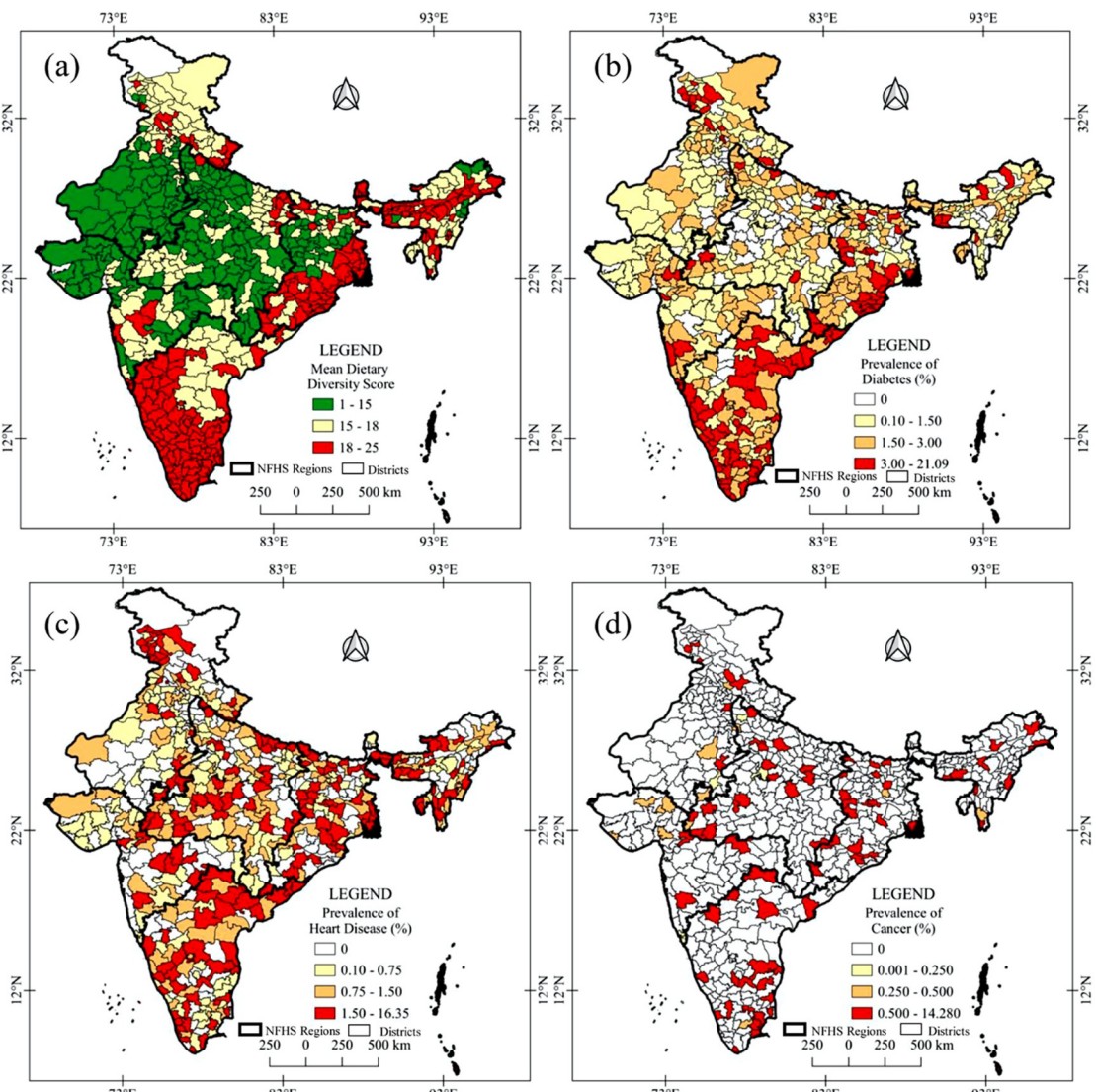

**Fig 2.** Geospatial pattern of (a) mean dietary diversity, the prevalence of (b) diabetes, (c) heart diseases, and (d) cancer among men in India, 2015–16. *(The source of the base map shapefiles available on DHS website at: https://spatialdata.dhsprogram.com. The data are freely available for acadèmic use. QGIS 3.14 software has been used to produce spatial distribution map of dietery diversity and non-communicable chronic diseases).*

diseases. Adults with medium dietary diversity scores show the lowest prevalence for coronary heart diseases (1.1 percent) and cancer (0.25 percent) than those with high and low dietary diversity categories. However, the dietary diversity score is positively associated with the prevalence of diabetes (p<0.05).

## 3.5. Multivariate analysis

The multivariate binary logistics regression model (Table 5) shows that dietary diversity score (DDS) is a significant predictor for the odds ratio of non-communicable diseases after controlling with other background characteristics. The odds of the occurrence of diabetes are higher (OR 2.1; 95% CL 1.7 to 2.6); p<0.05) for adults who have high dietary diversity scores (DDS) than those having low DDS. In reference to the low DDS category, adults with a medium level

**Table 2.** Prevalence of non-communicable diseases (diabetes, heart diseases, and cancer) among adult men (15–54 years) in India by selected background characterizes.

| Background Characteristics | Diabetes | | | Heart Disease | | | Cancer | | | Total |
|---|---|---|---|---|---|---|---|---|---|---|
| | Prevalence (%) | Numbers | $\chi^2$ | Prevalence (%) | Numbers | $\chi^2$ | Prevalence (%) | Numbers | $\chi^2$ | N |
| Age | | | | | | | | | | |
| 15–29 | 0.52 | 270 | $p<0.05$ | 0.59 | 304 | $p<0.05$ | 0.22 | 114 | $p<0.05$ | 51535 |
| 30–44 | 2.24 | 905 | | 1.29 | 523 | | 0.30 | 123 | | 40424 |
| 45–54 | 6.06 | 1221 | | 2.46 | 497 | | 0.38 | 77 | | 20162 |
| Marital Status | | | | | | | | | | |
| Unmarried | 0.63 | 249 | $p<0.05$ | 0.67 | 266 | $p<0.05$ | 0.25 | 98 | ns | 39762 |
| Married | 2.96 | 2097 | | 1.44 | 1022 | | 0.30 | 213 | | 70781 |
| Separated/Widow/Divorced | 3.19 | 50 | | 2.31 | 36 | | 0.24 | 4 | | 1578 |
| Education | | | | | | | | | | |
| No Education | 1.85 | 270 | $p<0.05$ | 1.66 | 242 | $p<0.05$ | 0.27 | 40 | ns | 14590 |
| Primary | 2.27 | 320 | | 1.62 | 229 | | 0.32 | 46 | | 14091 |
| Secondary | 2.05 | 1312 | | 1.05 | 675 | | 0.26 | 164 | | 64010 |
| Higher | 2.55 | 495 | | 0.92 | 178 | | 0.34 | 65 | | 19431 |
| Occupation | | | | | | | | | | |
| No occupation | 1.41 | 347 | $p<0.05$ | 0.86 | 211 | $p<0.05$ | 0.32 | 78 | ns | 24623 |
| Agricultural | 1.73 | 523 | | 1.41 | 427 | | 0.26 | 78 | | 30202 |
| Services | 2.99 | 239 | | 1.38 | 110 | | 0.54 | 43 | | 7991 |
| Skilled and unskilled manual | 2.07 | 599 | | 1.15 | 332 | | 0.24 | 49 | | 28911 |
| Others | 3.37 | 687 | | 1.19 | 243 | | 0.22 | 46 | | 20394 |
| Smoking | | | | | | | | | | |
| No | 1.98 | 1633 | $p<0.05$ | 0.98 | 808 | $p<0.05$ | 0.26 | 212 | ns | 82531 |
| Yes | 2.58 | 763 | | 1.74 | 516 | | 0.35 | 102 | | 29591 |
| Drinks alcohol | | | | | | | | | | |
| No | 1.84 | 1456 | $p<0.05$ | 0.97 | 765 | $p<0.05$ | 0.24 | 190 | $p<0.05$ | 79036 |
| Yes | 2.84 | 940 | | 1.69 | 559 | | 0.38 | 124 | | 33086 |
| Religion | | | | | | | | | | |
| Hindu | 2.08 | 1899 | $p<0.05$ | 1.17 | 1065 | $p<0.05$ | 0.29 | 261 | $p<0.05$ | 91390 |
| Muslim | 2.06 | 305 | | 1.06 | 157 | | 0.18 | 26 | | 14790 |
| Others | 3.23 | 192 | | 1.72 | 102 | | 0.45 | 27 | | 5942 |
| Caste | | | | | | | | | | |
| SC | 1.91 | 424 | $p<0.05$ | 1.22 | 271 | ns | 0.33 | 72 | $p<0.05$ | 22139 |
| ST | 1.17 | 116 | | 1.50 | 148 | | 0.20 | 20 | | 9872 |
| OBC | 2.20 | 1075 | | 1.17 | 571 | | 0.35 | 172 | | 48841 |
| General | 2.50 | 782 | | 1.07 | 335 | | 0.16 | 50 | | 31270 |
| Residence | | | | | | | | | | |
| Urban | 2.71 | 1163 | $p<0.05$ | 1.12 | 480 | ns | 0.29 | 123 | ns | 42952 |
| Rural | 1.78 | 1233 | | 1.22 | 844 | | 0.28 | 191 | | 69169 |
| Wealth Index | | | | | | | | | | |
| Poorest | 1.11 | 183 | $p<0.05$ | 1.47 | 242 | $p<0.05$ | 0.33 | 54 | $p<0.05$ | 16440 |
| Poorer | 1.22 | 255 | | 1.20 | 251 | | 0.20 | 41 | | 20904 |
| Middle | 1.82 | 430 | | 1.21 | 287 | | 0.34 | 79 | | 23687 |
| Richer | 2.60 | 648 | | 1.31 | 327 | | 0.34 | 85 | | 24976 |
| Richest | 3.37 | 880 | | 0.82 | 215 | | 0.21 | 55 | | 26114 |
| Region | | | | | | | | | | |
| North | 2.12 | 362 | $p<0.05$ | 1.31 | 223 | $p<0.05$ | 0.13 | 22 | $p<0.05$ | 117054 |

(*Continued*)

**Table 2.** (Continued)

| Background Characteristics | Diabetes | | | Heart Disease | | | Cancer | | | Total |
|---|---|---|---|---|---|---|---|---|---|---|
| | Prevalence (%) | Numbers | $\chi^2$ | Prevalence (%) | Numbers | $\chi^2$ | Prevalence (%) | Numbers | $\chi^2$ | N |
| Central | 3.22 | 18 | | 0.64 | 4 | | 0.20 | 1 | | 545 |
| East | 1.45 | 267 | | 0.87 | 160 | | 0.14 | 25 | | 18401 |
| Northeast | 2.48 | 641 | | 1.10 | 285 | | 0.19 | 50 | | 25819 |
| West | 2.72 | 147 | | 0.83 | 45 | | 0.12 | 6 | | 5416 |
| South | 2.14 | 961 | | 1.35 | 607 | | 0.47 | 209 | | 44887 |
| Total | 2.14 | 2396 | | 1.18 | 1324 | | 0.28 | 314 | | 112122 |

Note: SC-Scheduled caste; ST-Schedule tribes; OBC-Other backward caste; $\chi^2$-Chi-square test, *ns*- not significant.

of DDS are less likely to suffer from coronary heart disease (OR 0.88; 95% CL 0.7 to 1.0; p<0.10), and cancer (OR 0.71; 95% CL 0.5 to 0.9; p<0.05). Age, marital status, level of education, occupation status, drinking habits, and wealth index category is statistically significant and associated with the prevalence of diabetes, coronary heart diseases, and cancer. The odds of occurrence of diabetes are higher for married (OR 1.5; 95% CL 1.2 to 1.7; p<0.05) adult men compared to unmarried. In terms of occupation, the odds of diabetes are nearly half for the adults engaged in agriculture (OR 0.5; 95% CL 0.4 to 0.6; p<0.05) and manual labor (OR 0.6; 95% CL 0.5 to 0.7; p<0.05) compared to no occupation category. Similar associations are also observed in the case of coronary heart disease and cancer. Adults who drink alcohol are more likely to suffer from diabetes (OR 1.2; 95% CL 1.0 to 1.2; p<0.05), coronary heart disease (OR 1.3; 95% CL 1.2 to 1.5; p<0.05), and cancer (OR 1.5; 95% CL 1.1 to 1.9; p<0.05) in comparison to those who don't drink. People of more affluent wealth index categories are significantly more likely to suffer from diabetes than people in the poorest wealth index. However, the odds of coronary heart disease and cancer occurrence are significantly less in the upper wealth index categories than in the lower wealth index category.

## 4. Discussions

A healthy and optimum diet is essential for an active and healthy life. Therefore, Sustainable Development Goal (SDG) -2 also prioritizes diet diversity to ensure food and nutrition security. In this context, the present study aims to examine diet diversity and associations with non-communicable diseases among adult men in India. The study's findings are relevant in the Indian context, where the food availability, food selection, and food habits of the people are determined by a set of complex socio-economic, regional, and cultural factors [35–40]. Therewith, the rising prevalence of non-communicable diseases (NCDs) is also a growing concern. Anticipated the burden of NCDs in India, several studies have highlighted heterogeneity in the prevalence of cancer [41–43] and diabetes [44–47] across different socio-demographic groups and regions in India.

The present study shows that nearly half of adult men consume milk/curd, pulses/beans, and vegetables daily. However, consumption of protein-rich foods such as eggs, fish, and chicken is mostly occasional. Diet diversity also varies geographically across districts in India. The lowest diet diversity is found in the western part of India, whereas the southern-costal and eastern-costal parts of India have reported higher diet diversity (as in Fig 2A). Several factors, such as geographical availability of foods [48–52], accessibility to markets, purchasing capacity [53,54], level of education, individual tests, and preferences [55], etc., are the significant determinants of dietary diversity at the regional level in India [56,57].

**Table 3. Prevalence of non-communicable diseases (diabetes, heart diseases, and cancer) among adult men (15–54 years) in India by food consumption patterns.**

| Food Groups | Diabetes | | | Heart Disease | | | Cancer | | | |
|---|---|---|---|---|---|---|---|---|---|---|
| | Prevalence (%) | Numbers | $\chi^2$ | Prevalence (%) | Numbers | $\chi^2$ | Prevalence (%) | Numbers | $\chi^2$ | N |
| Milk or curd | | | | | | | | | | |
| Daily | 2.35 | 1218 | $p<0.05$ | 1.00 | 513 | $p<0.05$ | 0.29 | 152 | $p<0.05$ | 51863 |
| Weekly | 1.90 | 612 | | 1.13 | 364 | | 0.22 | 70 | | 32233 |
| Occasionally | 1.89 | 423 | | 1.41 | 316 | | 0.30 | 67 | | 22368 |
| Never | 2.52 | 143 | | 2.31 | 130 | | 0.44 | 25 | | 5658 |
| Pulses or beans | | | | | | | | | | |
| Daily | 2.27 | 1187 | $p<0.05$ | 1.20 | 628 | $p<0.05$ | 0.26 | 136 | $p<0.05$ | 52310 |
| Weekly | 1.84 | 906 | | 1.03 | 510 | | 0.20 | 98 | | 49362 |
| Occasionally | 2.87 | 287 | | 1.64 | 164 | | 0.73 | 73 | | 9985 |
| Never | 3.41 | 16 | | 4.63 | 21 | | 1.44 | 1 | | 464 |
| Dark green leafy vegetable | | | | | | | | | | |
| Daily | 2.28 | 1195 | $p<0.05$ | 1.34 | 701 | $p<0.05$ | 0.33 | 175 | $p<0.05$ | 52448 |
| Weekly | 1.97 | 918 | | 1.01 | 469 | | 0.19 | 86 | | 46530 |
| Occasionally | 2.16 | 273 | | 1.08 | 137 | | 0.39 | 49 | | 12641 |
| Never | 2.06 | 10 | | 3.22 | 16 | | 0.54 | 3 | | 503 |
| Fruits | | | | | | | | | | |
| Daily | 3.61 | 443 | $p<0.05$ | 1.52 | 186 | $p<0.05$ | 0.71 | 87 | $p<0.05$ | 12293 |
| Weekly | 2.04 | 902 | | 1.04 | 459 | | 0.22 | 98 | | 44285 |
| Occasionally | 1.83 | 977 | | 1.19 | 633 | | 0.24 | 126 | | 53377 |
| Never | 3.43 | 74 | | 2.08 | 45 | | 0.13 | 3 | | 2167 |
| Eggs | | | | | | | | | | |
| Daily | 3.63 | 198 | $p<0.05$ | 2.20 | 120 | $p<0.05$ | 1.19 | 65 | $p<0.05$ | 5441 |
| Weekly | 2.34 | 1171 | | 1.12 | 562 | | 0.18 | 88 | | 50166 |
| Occasionally | 1.88 | 645 | | 1.28 | 436 | | 0.38 | 128 | | 34218 |
| Never | 1.71 | 382 | | 1.00 | 205 | | 0.15 | 33 | | 22296 |
| Fish | | | | | | | | | | |
| Daily | 3.97 | 216 | $p<0.05$ | 2.34 | 127 | $p<0.05$ | 0.57 | 31 | $p<0.05$ | 5432 |
| Weekly | 2.57 | 978 | | 1.33 | 504 | | 0.28 | 106 | | 38062 |
| Occasionally | 1.97 | 751 | | 1.22 | 466 | | 0.36 | 137 | | 38176 |
| Never | 1.48 | 451 | | 0.74 | 226 | | 0.13 | 40 | | 30451 |
| Chicken or meat | | | | | | | | | | |
| Daily | 3.10 | 59 | $p<0.05$ | 1.82 | 35 | $p<0.05$ | 0.51 | 10 | $p<0.05$ | 1915 |
| Weekly | 2.67 | 1165 | | 1.27 | 553 | | 0.32 | 137 | | 43680 |
| Occasionally | 1.88 | 764 | | 1.28 | 520 | | 0.31 | 127 | | 40594 |
| Never | 1.57 | 407 | | 0.83 | 216 | | 0.15 | 39 | | 25933 |
| Fried food | | | | | | | | | | |
| Daily | 2.32 | 255 | $p<0.05$ | 1.30 | 137 | $p<0.05$ | 0.20 | 22 | $p<0.05$ | 10998 |
| Weekly | 1.96 | 774 | | 1.12 | 143 | | 0.26 | 102 | | 39613 |
| Occasionally | 2.06 | 1087 | | 1.14 | 442 | | 0.31 | 161 | | 52819 |
| Never | 3.21 | 279 | | 1.57 | 137 | | 0.33 | 29 | | 8690 |
| Aerated drinks | | | $p<0.05$ | | | $p<0.05$ | | | ns | |
| Daily | 2.32 | 166 | | 1.23 | 88 | | 0.20 | 14 | | 7180 |
| Weekly | 1.82 | 513 | | 0.90 | 256 | | 0.28 | 78 | | 28245 |
| Occasionally | 1.96 | 1234 | | 1.22 | 765 | | 0.30 | 187 | | 62826 |
| Never | 3.48 | 482 | | 1.55 | 215 | | 0.25 | 35 | | 13870 |

*(Continued)*

**Table 3.** (Continued)

| Food Groups | Diabetes | | | Heart Disease | | | Cancer | | | |
|---|---|---|---|---|---|---|---|---|---|---|
| | Prevalence (%) | Numbers | $\chi^2$ | Prevalence (%) | Numbers | $\chi^2$ | Prevalence (%) | Numbers | $\chi^2$ | N |
| Total | 2.14 | 2396 | | 1.18 | 1324 | | 0.28 | 314 | | 112121 |

$\chi^2$-Chi-square test; *ns*- not significant; Food group consumption frequency is based on 30 days recall period.

The study highlights that types of food and frequency of food consumption are significantly associated with the prevalence of NCDs. Optimum consumption (i.e., weekly and occasional) of selected foods such as milk/curd, pulses/beans, vegetables, etc., shows a lower prevalence of diabetes. Contrarily, daily or never-consumptions of these food groups are associated with a higher prevalence of diabetes. Studies have linked that optimum and balanced nutrition in the diet is essential to prevent diabetes [58]. On the other hand, daily consumption of fish, eggs, and chicken is positively associated with diabetes, heart disease, and cancer. Similar associations were also reported in many pieces of literature across the geographical regions of China, Sri Lanka, Tanzania, etc. [27,59,60], and India [27–34]. A recent study by Agarwal *et. al.*, (2014) has reported that daily fish intake is positively associated with diabetes among adult men and women in India [61]. Our study also shows that the prevalence of diabetes (3.9 percent), heart disease (2.3 percent), and cancer (0.6 percent) is highest among those adults who consumed fish daily than any other categories. Although, there was a study by Bharati *et el.* (2011) that could not establish any relationship between these two [62]. In consistence with other studies [44,58,62–65], the present study also shows that the risks of NCDs increase with age and drinking habits among adult men. Physical activities are critical in preventing NCDs. The study shows that adult men engaged in heavy activities, such as agriculture and manual work, are less likely to suffer from non-communicable diseases. In addition, a few socio-demographic characteristics such as education, caste, and wealth index are also significant in NCDs prevalence. Concerning the geographical risk factors of NCDs, the likelihood of diabetes is less in central India (OR 0.6; p<0.05) and more in the northeast (OR 1.2; p<0.05) and western parts of India (OR 1.4; p<0.05) compared to the northern part of India (OR 1). Similarly, heart disease and cancer are likely more prevalent in the northeast, western, and southern parts of India than in the northern part of India, where fish, chicken, and other protein-rich foods are higher [61,66].

## 5. Conclusion

Our study outlined a significant relationship between dietary diversity and the prevalence of non-communicable diseases among adult men in India. Both high and less-diversified foods in the diet have significant positive associations with the prevalence of NCDs. Therefore,

**Table 4. Association between dietary diversity score (DDS) and non-communicable diseases among adult men (15–54 years) in India.**

| Dietary Diversity Score (DDS) | Diabetes | | | Heart Disease | | | Cancer | | | |
|---|---|---|---|---|---|---|---|---|---|---|
| | Prevalence (%) | Numbers | $\chi^2$ | Prevalence (%) | Numbers | $\chi^2$ | Prevalence (%) | Numbers | $\chi^2$ | N |
| Low | 1.71 | 380 | *p*<0.05 | 1.25 | 277 | *p*<0.05 | 0.31 | 68 | *p*<0.05 | 22254 |
| Medium | 2.14 | 1835 | | 1.12 | 962 | | 0.25 | 210 | | 85657 |
| High | 4.31 | 181 | | 2.02 | 85 | | 0.86 | 36 | | 4210 |
| Total | 2.14 | 2396 | | 1.18 | 1324 | | 0.28 | 314 | | 112122 |

$\chi^2$-Chi-square test.

**Table 5. Multivariate binary logistic regression to show the odds ratio (OR) of non-communicable diseases (diabetes, heart diseases, and cancer) among adult men (15–54 years) in India.**

| Background Characteristics | Diabetes | | Heart Disease | | Cancer | |
|---|---|---|---|---|---|---|
| | OR | *p*-value | OR | *p*-value | OR | p-value |
| Dietary Diversity Score (DDS) | | | | | | |
| Low | 1 | . | 1 | . | 1 | . |
| Medium | 1.092 [0.969–1.229] | - | 0.881 [0.771–1.006] | p<0.10 | 0.715 [0.543–0.941] | p<0.05 |
| High | 2.153 [1.769–2.621] | p<0.05 | 1.267 [0.951–1.688] | - | 1.461 [0.847–2.521] | - |
| Age | | | | | | |
| 15–29 | 1 | . | 1 | . | 1 | . |
| 30–44 | 3.082 [2.584–3.676] | p<0.05 | 2.036 [1.689–2.455] | p<0.05 | 1.614 [1.122–2.322] | p<0.05 |
| 45–54 | 8.397 [7.008–10.063] | p<0.05 | 3.93 [3.224–4.789] | p<0.05 | 1.772 [1.158–2.712] | p<0.05 |
| Marital Status | | | | | | |
| Unmarried | 1 | . | 1 | . | 1 | . |
| Married | 1.47 [1.224–1.766] | p<0.05 | 1.086 [0.891–1.324] | - | 0.99 [0.675–1.452] | - |
| Separated/Widow/Divorced | 1.164 [0.79–1.716] | - | 1.305 [0.881–1.934] | - | 0.818 [0.286–2.339] | - |
| Education | | | | | | |
| No Education | 1 | . | 1 | . | 1 | . |
| Primary | 1.248 [1.039–1.498] | p<0.05 | 1.103 [0.91–1.338] | | 0.972 [0.597–1.581] | - |
| Secondary | 1.501 [1.284–1.754] | p<0.05 | 1.189 [1.007–1.404] | p<0.05 | 1.412 [0.956–2.088] | p<0.10 |
| Higher | 1.702 [1.409–2.055] | p<0.05 | 1.242 [0.985–1.565] | p<0.10 | 1.735 [1.05–2.868] | p<0.05 |
| Occupation | | | | | | |
| No occupation | 1 | . | 1 | . | 1 | . |
| Agricultural | 0.519 [0.443–0.609] | p<0.05 | 0.858 [0.709–1.039] | - | 0.704 [0.479–1.035] | p<0.10 |
| services | 0.886 [0.741–1.059] | - | 0.899 [0.699–1.158] | - | 1.004 [0.612–1.646] | - |
| Skilled and unskilled manual | 0.632 [0.541–0.738] | p<0.05 | 0.775 [0.637–0.942] | p<0.05 | 0.615 [0.413–0.915] | p<0.05 |
| Others | 0.792 [0.679–0.923] | p<0.05 | 1.062 [0.866–1.303] | - | 0.924 [0.609–1.401] | - |
| Smoking | | | | | | |
| No | 1 | . | 1 | . | 1 | . |
| Yes | 1.002 [0.907–1.107] | - | 1.251 [1.11–1.41] | p<0.05 | 0.864 [0.651–1.147] | - |
| Drinks alcohol | | | | | | |
| No | 1 | . | 1 | . | 1 | . |
| Yes | 1.189 [1.077–1.312] | p<0.05 | 1.318 [1.164–1.493] | p<0.05 | 1.462 [1.118–1.912] | p<0.05 |
| Religion | | | | | | |
| Hindu | 1 | . | 1 | . | 1 | . |
| Muslim | 1.151 [1.004–1.321] | p<0.05 | 1.502 [1.274–1.772] | p<0.05 | 0.693 [0.431–1.113] | - |
| Others | 1.287 [1.113–1.489] | p<0.05 | 1.142 [0.942–1.384] | - | 1.445 [0.959–2.176] | p<0.10 |
| Caste | | | | | | |
| SC | 1 | . | 1 | . | 1 | . |
| ST | 0.781 [0.657–0.929] | p<0.05 | 0.872 [0.716–1.062] | - | 0.547 [0.353–0.848] | p<0.05 |
| OBC | 0.919 [0.806–1.047] | - | 0.889 [0.759–1.042] | - | 1.037 [0.767–1.402] | - |
| General | 0.996 [0.866–1.145] | - | 0.957 [0.801–1.142] | - | 0.507 [0.334–0.77] | p<0.05 |
| Residence | | | | | | |
| Urban | 1 | . | 1 | . | 1 | . |
| Rural | 1.076 [0.969–1.195] | - | 0.994 [0.863–1.144] | - | 0.979 [0.722–1.328] | - |
| Wealth Index | | | | | | |
| Poorest | 1 | . | 1 | . | 1 | . |
| Poorer | 1.001 [0.831–1.205] | - | 0.967 [0.812–1.151] | - | 0.579 [0.4–0.838] | p<0.05 |
| Middle | 1.208 [1.007–1.448] | p<0.05 | 0.793 [0.658–0.956] | p<0.05 | 0.518 [0.354–0.759] | p<0.05 |

(*Continued*)

**Table 5.** (Continued)

| Background Characteristics | Diabetes | | Heart Disease | | Cancer | |
|---|---|---|---|---|---|---|
| | OR | *p*-value | OR | *p*-value | OR | p-value |
| Richer | 1.479 [1.23–1.779] | p<0.05 | 0.785 [0.641–0.96] | p<0.05 | 0.499 [0.331–0.751] | p<0.05 |
| Richest | 1.776 [1.458–2.163] | p<0.05 | 0.596 [0.471–0.755] | p<0.05 | 0.318 [0.194–0.522] | p<0.05 |
| Region | | | | | | |
| North | 1 | . | 1 | . | 1 | . |
| Central | 0.6 [0.432–0.834] | p<0.05 | 0.809 [0.524–1.251] | - | 1.357 [0.456–4.041] | - |
| East | 0.972 [0.825–1.145] | - | 1.138 [0.933–1.389] | - | 1.447 [0.836–2.504] | - |
| Northeast | 1.171 [1.015–1.352] | p<0.05 | 1.367 [1.143–1.635] | p<0.05 | 2.971 [1.851–4.769] | p<0.05 |
| West | 1.448 [1.207–1.737] | p<0.05 | 1.222 [0.954–1.565] | - | 2.848 [1.623–4.999] | p<0.05 |
| South | 0.983 [0.852–1.134] | - | 1.189 [0.998–1.418] | p<0.10 | 2.69 [1.703–4.25] | p<0.05 |
| Constant | 0.003 [0.002–0.004] | p<0.05 | 0.005 [0.004–0.008] | p<0.05 | 0.002 [0.001–0.004] | p<0.05 |
| Pseudo r-squared | 0.094 | | 0.036 | | 0.033 | |

Note: SC-Scheduled caste; ST-Schedule tribes; OBC-Other backward caste.

optimum and moderate diet diversity is recommended to prevent these NCDs. The study suggests that frequent or daily consumption of high-protein-rich foods such as fish, eggs, and chicken in the diet can increase the risks of coronary heart disease and cancer. Therefore, we stress the importance of quality diet diversity as an integrated component of food and nutrition security as defined by the World Food Summit (1996). In this context, it is essential to enhance the knowledge and awareness about the nutritional values of foods and food choices to prevent non-communicable diseases among men in India. In addition to diet diversity patterns, the study also signifies the importance of selected socio-demographic characteristics, such as, age, educational status, occupation, drinking habits, caste, and wealth index, as predictors of NCDs among adult men in India.

Few limitations needed to be acknowledged to interpret the results. First, the present study utilized cross-sectional data, which may not truly confirm the cause-and-effect associations between dietary diversity and non-communicable diseases of individuals. The case and control study could have been more appropriate than a cross-sectional study. Second, the study is limited to the male adult population only; hence, can not be generalized for the whole population of the country. Third, there is likely a recall bias among the respondents answering the questions on food consumption in the last 30 days. Further, the data does not have any nutrient-specific information, i.e., calorie, protein, fat, carbohydrates, etc., rather, it uses various food groups and their consumption frequencies as a proxy of nutrient intakes. Fourth, the study could not incorporate the contribution of other variables, such as weight, exposure to physical activities, etc., which are highly relevant in contributing to the prevalence of NCDs

## Author Contributions

**Conceptualization:** Sanjit Sarkar.

**Data curation:** Mriganka Dolui, Pritam Ghosh.

**Formal analysis:** Mriganka Dolui, Pritam Ghosh.

**Investigation:** Pritam Ghosh.

**Methodology:** Sanjit Sarkar.

**Software:** Mriganka Dolui, Sanjit Sarkar.

**Supervision:** Sanjit Sarkar, Moslem Hossain.

**Validation:** Sanjit Sarkar.

**Writing – original draft:** Sanjit Sarkar.

**Writing – review & editing:** Moslem Hossain.

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
