## [Decision Letter · Decision Letter 0]

10 Jan 2023

PGPH-D-22-01943

Dietary Diversity and Association with Non-communicable Diseases (NCDs) among Adult Men (15-54 Years): A Cross-Sectional Study using National Family and Health Survey, India

Dear Dr. Sarkar,

Thank you for submitting your manuscript to PLOS Global Public Health. After careful consideration, we feel that it has merit but does not fully meet PLOS Global Public Health’s publication criteria as it currently stands. Therefore, we invite you to submit a revised version of the manuscript that addresses the points raised during the review process.

We look forward to receiving your revised manuscript.

Kind regards,

Rajesh Sharma, Ph.D.

Academic Editor

Journal Requirements:

2. Please provide separate figure files in .tif or .eps format only and remove any figures embedded in your manuscript file. Please also ensure that all files are under our size limit of 10MB.

3. We do not publish any copyright or trademark symbols that usually accompany proprietary names, eg  ©, ®, ™  (e.g. next to drug or reagent names). Please remove all instances of trademark/copyright symbols throughout the text, including ® on pages 24 and 25.

Additional Editor Comments (if provided):

Minor Revision

Reviewers' comments:

Reviewer's Responses to Questions

**Comments to the Author**

1. Does this manuscript meet PLOS Global Public Health’s publication criteria? Is the manuscript technically sound, and do the data support the conclusions? The manuscript must describe methodologically and ethically rigorous research with conclusions that are appropriately drawn based on the data presented.

Reviewer #1: Yes

Reviewer #2: Yes

2. Has the statistical analysis been performed appropriately and rigorously?

Reviewer #1: I don't know

Reviewer #2: Yes

3. Have the authors made all data underlying the findings in their manuscript fully available (please refer to the Data Availability Statement at the start of the manuscript PDF file)?

Reviewer #1: Yes

Reviewer #2: Yes

4. Is the manuscript presented in an intelligible fashion and written in standard English?

Reviewer #1: Yes

Reviewer #2: No

5. Review Comments to the Author

Reviewer #1: 1/ Introduction need to be shortened. Certain parts can be move to discusison

2/ Discussion, despite moving parts from introduction, still need to be concise

3/ Any reference to Dietary Diversity Score? Was based on prior work? Please clarify. Also a more clear description in introduction may be valuable as part of aims of the study

4/ Please clarify why specific interest in males, why not females, and what data that may require an independent study?

5/ Considering the multivariate analyses, suggest looking into a Bayesian model analysis

Reviewer #2: Reference for 'three-quarters of global NCD deaths occur in low- and middle-income

countries.'

'Noncommunicating diseases'- non-communicable

It would be more appropriate to replace 'heart disease' with coronary heart disease

'optimal growth' should be omitted

Explaining the process and reasoning for developing the DDS will help validate the system. As there is a confusion if the authors are trying to validate the frequency of consumption of certain food groups, such as linking a high consumption of protein to diabetes or linking 'diversity' of different food groups to NCDs

Why was weight category not included in the co-variates? As obesity is a comorbidity

Why weren't carbohydrates included in the food groups? Since previous studies have linked a high intake of carbohydrates, particularly processed, to a higher incidence of diabetes

A limitation of the study would be to mention that the 'quantities of protein consumed' were not included rather only frequency 'Our study also shows that the prevalence of diabetes (3.9 percent), heart disease (2.3 percent), and cancer (0.6 percent) is highest among those adults who consumed fish daily than any other category.'

The study is quite general and reads a bit distracting

6. PLOS authors have the option to publish the peer review history of their article (what does this mean?). If published, this will include your full peer review and any attached files.

**Do you want your identity to be public for this peer review?** For information about this choice, including consent withdrawal, please see our Privacy Policy.

Reviewer #1: No

Reviewer #2: **Yes: **Dr. Samaa Akhtar

---

## [Decision Letter · Decision Letter 1]

13 Mar 2023

Dietary Diversity and Association with Non-communicable Diseases (NCDs) among Adult Men (15-54 Years): A Cross-Sectional Study using National Family and Health Survey, India

PGPH-D-22-01943R1

Dear Dr. Sarkar,

We are pleased to inform you that your manuscript 'Dietary Diversity and Association with Non-communicable Diseases (NCDs) among Adult Men (15-54 Years): A Cross-Sectional Study using National Family and Health Survey, India' has been provisionally accepted for publication in PLOS Global Public Health.

Best regards,

Rajesh Sharma, Ph.D.

Academic Editor

Reviewer Comments (if any, and for reference):

Reviewer's Responses to Questions

**Comments to the Author**

1. If the authors have adequately addressed your comments raised in a previous round of review and you feel that this manuscript is now acceptable for publication, you may indicate that here to bypass the “Comments to the Author” section, enter your conflict of interest statement in the “Confidential to Editor” section, and submit your "Accept" recommendation.

Reviewer #2: All comments have been addressed

2. Does this manuscript meet PLOS Global Public Health’s publication criteria? Is the manuscript technically sound, and do the data support the conclusions? The manuscript must describe methodologically and ethically rigorous research with conclusions that are appropriately drawn based on the data presented.

Reviewer #2: Yes

3. Has the statistical analysis been performed appropriately and rigorously?

Reviewer #2: Yes

4. Have the authors made all data underlying the findings in their manuscript fully available (please refer to the Data Availability Statement at the start of the manuscript PDF file)?

Reviewer #2: Yes

5. Is the manuscript presented in an intelligible fashion and written in standard English?

Reviewer #2: No

6. Review Comments to the Author

Reviewer #2: Well done. The concept, methods and results are cohesive and scientifically sound. If I would nit pick, there are some issues with grammar. However, the manuscript can be understood regardless.

7. PLOS authors have the option to publish the peer review history of their article (what does this mean?). If published, this will include your full peer review and any attached files.

**Do you want your identity to be public for this peer review?** For information about this choice, including consent withdrawal, please see our Privacy Policy.

Reviewer #2: **Yes: **Dr Samaa Akhtar
